# The Role of Thromboprophylaxis in Patients with Portal Vein Thrombosis: A Life-Threatening Complication after Laparoscopic Sleeve Gastrectomy Following 8 Years of Experience in a Bariatric Center of Excellence

**DOI:** 10.3390/diagnostics13010043

**Published:** 2022-12-23

**Authors:** Daniela Godoroja-Diarto, Catalin Copaescu, Elena Rusu, Alina Constantin

**Affiliations:** 1Department Anaesthesia and Intennsive Care, Ponderas Academic Hospital, 014142 Bucharest, Romania; 2Department of Surgery, Ponderas Academic Hospital, 014142 Bucharest, Romania; 3Faculty of Medicine, University Titu Maiorescu, 040441 Bucharest, Romania; 4Department Gastroenterology, Ponderas Academic Hospital, 014142 Bucharest, Romania

**Keywords:** portal vein thrombosis, mesenteric vein thrombosis, laparoscopic sleeve gastrectomy

## Abstract

Porto-mesenteric vein thrombosis (PVMT) is a rare but life-threatening complication after laparoscopic sleeve gastrectomy (LSG). Laparoscopic sleeve gastrectomy (LSG) is considered the most common procedure for efficiently realizing weight loss and treating obesity-related co-morbidities. This study aimed to shed light on this relatively rare complication by presenting a series of patients who developed PMVT after LSG in light of the need to change the specific protocol of thromboprophylaxis in bariatric patients. We proposed to answer two questions: whether we should perform a thrombophilia workup as a standard practice and whether we should extend chemoprophylaxis to more than 3 weeks among all bariatric patients. This study also aimed to investigate the possible risk factors and eventually present our updated protocol for PMVT management and prophylaxis.

## 1. Introduction

Porto-mesenteric vein thrombosis (PMVT) is an infrequent but potentially fatal complication after laparoscopic sleeve gastrectomy (LSG) and can result in gastrointestinal tract ischemia and small intestine infarction if the PMVT is occlusive [1,2,3,4,5].

Amongst the surgical options, laparoscopic sleeve gastrectomy (LSG) is considered the most common procedure, proven to be efficient for weight loss and obesity-related co-morbidities [4,6]. Although LSG is safe, with low morbidity, complications can occur; the most common complications are sepsis, bleeding, and pulmonary embolism [7]. 

All severely obese patients are at an increased risk of thromboembolic events and ischemia secondary to vessel wall damage and inflammatory condition with hypercoagulability [7,8]. PMVT is a rare complication, but it has been increasingly reported over the last 10 years, especially after LSG. The reported incidence of PMVT after LSG ranges from 0.3% to 3% [1,7,9].

Despite progress in laparoscopic techniques and enhanced recovery in bariatric surgery, venous thromboembolic complications are still a major concern. Due to the nonspecific clinical symptoms that make it difficult to diagnose PMVT, and its association with the potential for severe evolution, the best protocol for its prevention and treatment is still open to debate. Heparin anticoagulation is the most commonly used method to treat PMVT, even though heparin alone has an important failure rate—around 65% for non-surgical cases, according to some authors [10]. Our team has been challenged by several complex cases of PMVT, and, consequently, in 2011, we introduced a protocol for preventing venous thrombosis in the current bariatric surgery practice. The outcomes of this protocol were evaluated in 2012, with a PMVT rate of 0.19% [11]. The algorithm was further improved when our hospital became a Center of Excellence in Bariatric Surgery, and the protocol with adjusted doses of LMWH and extended prophylaxis of up to 21 days was applied to all bariatric patients. After a 5-year study from 2014 to 2019 on the incidence and outcomes of PMVT after LSG in our Center of Excellence in Bariatric Surgery, a diagnostic and therapeutic protocol for PMVT after LSG was proposed and applied. Even in these circumstances, due to the potentially fatal consequence of this complication, we still have concerns related to it. The incidence of hereditary thrombophilia in bariatric patients with PMVT is reported in the literature to be between 3% and 52% [6]. The timing of discovering PMVT varies as well, but the majority of the cases occur within the first month after the procedure [7]. Additionally, 50% of our PMVT cases developed after thromboprophylaxis was completed.

Our study aims to shed light on this relatively rare complication by presenting a series of patients who developed PMVT after LSG, as well as an up-to-date analysis of our database in light of the need to change our protocol for thromboprophylaxis in bariatric patients. We propose to answer two questions: should we perform a thrombophilia workup as a standard practice, and should we extend chemoprophylaxis by more than 3 weeks for all bariatric patients? This study also aims to explore possible risk factors and eventually present our updated protocol for PMVT management and prophylaxis.

## 2. Materials and Methods

Our study is an observational retrospective study. All the patients who underwent LSG between 1 November 2014 and 15 October 2022 in Ponderas Academic Hospital, a high-volume Center of Excellence for Bariatric and Metabolic Surgery, were reviewed in BOLD, the Bariatric Outcomes Longitudinal Database, where patients are prospectively registered as per SRC accreditation program requirements [12]. As an observational study, there was no intervention in medical protocols. We obtained institutional ethics committee approval for the study, and all the patients provided informed consent. We excluded from the study the patients who underwent any other primary or revisional bariatric procedure.

All patients with American Society of Anesthesiologists (ASA) physical status I–IV underwent a preoperative workup by a multidisciplinary team 1–4 weeks prior to surgery, which included a cardiologist and an anesthetist, who prescribed the DVT prophylaxis or perioperative anticoagulation regimen as per the protocol of our hospital. The protocol described below included perioperative LMWH and sequential compression devices intraoperatively.

All the bariatric patients received anesthesia with a standardized low-opioid protocol and a multimodal non-opioid postoperative analgesia [13].

The surgery was performed in reverse Trendelenburg, with carbon dioxide insufflation pressure of 12–15 mm Hg. The surgical protocol for LSG includes dissection of the posterior aspect of the fundus and an active search for the hiatal hernia, as well as calibration of the stomach with a 35 F bougie catheter for division that starts 1–2 cm proximal to the pylorus and ends 1 cm lateral to the angle of His. The stapled line was entirely over-sewn in all cases, followed by the methylene blue test; meanwhile, local hemostasis verified that the blood pressure was elevated by 30% compared to the preoperative status during specimen removal.

All the patients received a prophylactic antibiotic regimen with first-generation cephalosporines and postoperative fluids to maintain proper hydration. We recommended early ambulation at 6–8 h postoperative. Patients were discharged on the second postoperative day, depending on adequate clear liquid intake orally, thus avoiding the risk of dehydration.

### 2.1. Current Anticoagulation Venous Thrombosis Prophylaxis 

In 2011, we introduced a protocol for venous thrombo-prophylaxis, which we applied to all bariatric patients, irrespective of the risk, for 21 postoperative days in this study (Table 1).

The protocol was based on the recommendations of the UK Hemostasis, Anticoagulation, and Thrombosis (HAT) Committee, published on April 2010 [14,15], and used LMWH dose-adjusted to body weight. Our protocol was certified by the recent guidelines of the European Society of Anesthesiology [16], published in 2018, as well as the EAES guidelines, published in 2020 [17]. Some of the patients needed anticoagulation treatment with LMWH based on the specific recommendation of our cardiologist, 21 days postoperatively, followed by bridging to oral therapy when indicated. 

In January 2018, we introduced the measurement of anti-factor Xa (anti-Xa) [16] for monitoring both prophylactic or therapeutic doses of LMWH in difficult bariatric cases, especially in patients with severe obesity, thrombotic risk factors, or a history of DVT. We first evaluated the plasma levels of anti-Xa four hours after the third dose of LMWH. 

From our database, we extracted and analyzed the data of patients presented with PMVT after LSG during this period.

### 2.2. PMVT Diagnosis and Treatment

The protocol for the patients presented to the emergency room with the suspicion of PMVT includes screening for laboratory testing, DDimers, and evaluation of the PV and its mesenteric branches with abdominal Doppler ultrasound and thoraco-abdominal CT-angiograms using IV contrast, to determine the status of the portal and mesenteric venous thrombosis and investigate other potential sites of evolutive thrombosis.

The first method for treating all PMVT patients is intravenous heparin (bolus followed by continuous infusion) to achieve a therapeutic range of activated partial thromboplastin time (aPTT) 2–3 times baseline. The following steps are determined by the severity of disease, especially when an occlusive form of thrombosis occurs. Patients with occlusive disease can be treated with systemic tissue plasminogen activator (TPA) infusion followed by heparin. Surgery aims to remove the thrombi from the porto-mesenteric tree or to resect the ischemic bowel loops. In some challenging cases, bowel or spleen ischemia or necrosis can be diagnosed by laparoscopy.

After treating the acute stage of PMVT, the patients received oral anticoagulation with warfarin or a factor Xa inhibitor (rivaroxaban, apixaban) for at least six months, but most of the patients remained under long-term therapy.

### 2.3. Statistical Method

Statistical analysis was performed with SPSS version 22 (Chicago, IL, USA). Categorical data were reported as frequencies and percentages, and continuous data as average (mean) and standard deviation, after checking for normality (one-sample case Kolmogorov–Smirnov test). Only descriptive statistics were used.

## 3. Results

Between 1 November 2014 and 15 October 2022, 5154 patients underwent elective LSG at the Ponderas Academic Hospital, Bucharest, Romania, and their descriptions can be seen in Table 2.

Out of these, four patients were readmitted 7–60 days after LSG for PMVT (Table 3), equating to an incidence of 4/5154 (0.077%). Meanwhile, DVT incidence in the same period was 3/5154 (0.058%). Other postoperative complications in the cohort were bleeding fistula (10/5154) (0.19%) and sepsis (8/5154) (0.15%).

The characteristics of the PMVT group were the following:

Mean age: 40 years;

Average BMI: 46.85 kg/m^2^;

Average weight: 137 kg;

Average time of surgery for LSG: 104 min (range: 60–135);

Average days after LSG: 29;

Average hospital stay (days): 9.

The symptoms of the PMVT patients were nonspecific, such as malaise, nausea, abdominal or back pain, and fever. 

All patients had venous thrombosis risk factors, such as severe obesity, smoking behavior, and a personal or family DVT history.

The maximum intra-abdominal pressure used in all cases was 15 mm Hg. The diagnosis of the patients was made using Doppler ultrasound and CT (Figure 1, Occlusive PMVT). All patients underwent conservative therapy with intravenous heparin without surgical treatment, followed by lifelong anticoagulant therapy for the three survivors in the group. 

## 4. Discussion

PMVT is a rare but serious complication after bariatric surgery that is more common after LSG in the first postoperative month but has been increasingly reported in recent years as the number of laparoscopic bariatric operations continues to extend. Due to its life-threatening potential, all studies emphasize the necessity of a high index of clinical suspicion for prompt diagnosis and treatment.

Our interest in PVT incidence and outcomes started much earlier than the current and previous studies published in 2019 [18]. We analyzed a cohort of patients operated on between January 2008 and September 2012 [11]. During that period, we worked to develop and implement specific protocols for the purpose of obtaining accreditation as a Center of Excellence in Bariatric Surgery by SRC and IFSO-EC. In the studied period, we found that the incidence of PVT was 0.19% (5 out of 2220 patients with LSG), and all the patients had a good outcome [11]. 

The treatment of PMVT patients was with intravenous UFH in that series, except for a woman with a complete obstruction of the portal vein, who received systemic thrombolysis with intravenous TPA followed by heparin infusion. Her outcome was favorable, and she was further diagnosed with Leiden V factor thrombophilia. At the same time, we admitted one critical patient with PMVT and bowel necrosis operated on in another hospital. All these challenges were in the context of the first cases reported at the time, with little knowledge about the topic, which raised awareness of this underestimated, dangerous complication and helped us to understand that the outcome depends on early diagnosis and treatment.

In 2011, we also introduced the protocol of prophylaxis of thrombotic events with adjusted doses of LMWH for 21 days post-discharge, as described in the Materials and Methods section (Table 1).

To avoid dehydration, the patients were discharged when they could drink 1500 mL of clear fluids a day. Additionally, at the first sign of suspicion of PMVT, we applied all methods for early diagnosis and treatment. 

In our previous study [18], we described all three PMVT cases found between 2014 and 2019, including a dramatic case with an aggressive occlusive form of PMVT and a fatal outcome. The young patient had major thrombosis-risk-associated factors, and he was discharged after LSG with LMWH anticoagulation treatment as opposed to prophylactic anticoagulation treatment. Even though he presented on the eighth day with clinical symptoms of occlusive PMVT and shock, the dramatic evolution did not yield any room for intervention, except intravenous heparin, until his fatal outcome occurred [18].

The actual study thus aimed to evaluate the changes that could be applied to the protocol used before for the prophylaxis and treatment of venous thrombosis, especially PMVT, due to its potentially lethal outcomes.

In our present study analyzing all the cases of PMVT post-LSG from the last 8 years, although the PMVT incidence was very low (0.077%) and lower than in the literature [9], we noticed that 50% of the PMVT cases appeared after thromboprophylaxis was completed, and we were concerned about the severe cases that we encountered.

The first meta-analysis reported an incidence of PMVT of 0.4% based on 13 studies and 68 PMVT/16,237 operations [19].

Most of the studies show a higher PMVT incidence after LSG than from other bariatric operations. For example, two different, large retrospective studies that investigated PMVT in patients after bariatric surgery, by Goitein et al. (5706 patients) [7] and Rottenstreich et al. (4386 patients) [20], did not find PMVT after laparoscopic Roux-en-Y gastric bypass or biliary pancreatic diversion.

Although a precise explanation for the increased incidence of PMVT following LSG remains unclear, other meta-analyses [21] and studies suggest that this higher incidence can be explained by the following factors.

The prolonged use of a liver retractor may result in liver congestion and clot formation [22];The ligation of the gastroepiploic and short gastric vessels, resulting in gastric and splenic venous reflux near the splenic vein, could potentially initiate thrombosis, and the thermal effect of energy devices used for ligation can damage the splenic vein via mechanical or thermal effects [1,9,22,23];The increase in intra-abdominal pressure beyond 14 mmHg reduces the portal venous flow by 50% [10], which is decreased by the prolonged reverse-Trendelenburg position [24];Hypercapnia can reduce venous blood flow by vasospasm and increases the risk of thrombosis [25,26,27,28];Dehydration is a risk factor for thrombosis [28,29,30,31], especially in LSG patients, due to a reduction in gastric capacity; consequently, in the post-discharge summary for all the LSG patients, we recommend avoiding dehydration and exposure to heat for the first postoperative month and receiving intravenous fluids when they cannot drink 1 L of clear fluid.

Amongst the various risk factors, obesity is intrinsically associated with an increased risk for thrombotic events due to hypercoagulable states and the release of proinflammatory mediators, thus leading to a linear relationship between obesity and thromboembolic events. Other risk factors were found to include a personal history of malignancy or type 2 diabetes [32]. This was the reason that we applied the thromboprophylaxis protocol to all patients.

The authors of a meta-analysis [21] suggest the need to identify high-risk patients. 

The meta-analysis determined that the thrombophilia workup test was positive in 47.4% of patients who developed PMVT after LSG.

Another systematic review [6] looked at porto-mesenteric vein thrombosis after LSG and evaluated 28 studies consisting of 89 patients; the authors found thrombophilia present in at least 56% of PMVT patients. Parikh et al. assessed thrombophilia in all patients after 2018 (1075 patients) and found a rate of positive thrombophilia panel of 52.4% (563/1075), including FVIII elevation and antithrombin III deficiency protein S. FVIII elevation was the most common hematologic abnormality identified in PMVT (70%) in another retrospective study on 40 patients with PMVT after LSG; one third of them also experienced dehydration [33]. This suggests that FVIII elevation may be a precipitating factor for PMVT; therefore, it can be considered a screening test for PMVT [34].

Most of the literature is in the form of case series; however, in Shoar et al.’s [19] systematic review, half (52.4%) of the study population had a positive thrombophilia panel. Because of this high incidence of thrombophilia in bariatric patients, this study considered whether we should perform a thrombophilia workup as a standard practice. Unfortunately, we were challenged by the cost of the thrombophilia panel and patients being unwilling to undergo treatment. All our patients with PMVT were suspected for thrombophilia, but they did not wish to be screened for the panel; therefore, we could not diagnose it.

Genetic coagulation disorder screening for thrombophilia may be worth proposing, at least in high-risk patients, but its acceptance and cost remain issues.

The most common presentations of PMVT were not specific, with abdominal pain, nausea/vomiting, leukocytosis, and fever [1,7,9,35,36,37,38]. The severity of symptoms can vary; they are directly proportional to the extent of mesenteric venous thrombosis because of bowel ischemia [31] and can be fulminant with septic shock and organ failure, as described in our case. If the patient had not delayed his presentation to the hospital, an earlier diagnosis could have been made, and we could have had time to attempt thrombolysis, despite all of its risks.

This is why we encourage clinicians to maintain a high index of suspicion in cases of PMVT, allowing the early diagnosis and prompt management of this rare but life-threatening complication.

The literature suggests that symptoms appeared, on average, 12 to 15 days postoperatively [1,7,9,35]. However, in our cases, two symptoms (50%) presented after chemoprophylaxis ended.

The clinical diagnosis of PMVT should be confirmed by Doppler ultrasound and a CT scan with IV contrast [39]. Contrast CT is recommended as the first line, with a sensitivity of 90% [39,40], and it is very useful for recognizing PMVT evolution under treatment.

### 4.1. Treatment

Treatments ranged from unfractionated heparin UFH to bowel resection and liver transplantation. The complexity of the treatment methods depends on the timing of PMVT after bariatric surgery; the extent of the vascular involvement; and the severity of the ischemic damage of the bowel, spleen, or liver.

Anticoagulation: It has been recommended that subjects with acute PVT should be treated with anticoagulation, preferably UFH iv, as early as possible, which enhances recanalization [41,42]. All of our patients were treated with anticoagulation–heparin therapy, with good outcomes, except for the case with fulminant evolution. 

There is no consensus regarding the optimal duration and extent of anticoagulation. Ghandi et al. [10] recommended 3–6 months of anticoagulation; another study suggested a longer duration of up to 12 months [43]. However, patients with thrombophilia and extensive thrombosis must be on lifelong anticoagulation therapy [44], as was the case with all of our treated patients. Studies have shown that anticoagulation may result in recanalization in 48% of cases [36]. Recanalization occurred in our patients as well (Figure 2 and Figure 3).

Thrombolysis: Some studies recommend early thrombolysis to treat acute PMVT, suggesting that, after thrombolysis, there are higher rates of recanalization compared with heparin infusion [45,46,47]. These studies refer to patients with cirrhosis and malignancies, not postoperative patients.

One study reported successful thrombolysis for PMVT in a patient with ischemic bowel at laparotomy. After percutaneous transhepatic thrombolysis of the portal vein with a continuous infusion for 2 days, the second and third relook laparotomies showed a viable bowel without the need for surgical bowel resection [7,48,49].

We only experienced one case that we treated with systemic thrombolysis TPA due to severe occlusive PMVT in a young woman consequently diagnosed with Leyden factor thrombophilia, but before the period of the current study [11].

Several researchers have supported either percutaneous or transhepatic thrombolytic therapy when anticoagulation does not elicit a progressive response in severe presentations [1,7,35,50]. Superior mesenteric artery (SMA)-directed tissue plasminogen activator (t-PA) or thrombectomy may be superior in the acceleration of recanalization [50]. In the postoperative setting, thrombolytic therapy must be considered very carefully before being initiated, as there is a possibility of further surgery, with the potential for bowel resection. Further research in this area is recommended.

Surgery: The role of laparoscopic exploration in diagnosing PVT is still under debate. The decision regarding diagnostic laparoscopy should be based on the clinical evolution of the patient and CT findings [10] and be considered when the patient deteriorates and CT findings are not conclusive. No patients included in the present study required diagnostic laparoscopy. 

### 4.2. Prevention

To the best of our knowledge, there are two guidelines (ESAIC and EAES [16,17]) that refer to the prevention of thromboembolic events in bariatric patients, but they do not refer to the prevention of PMVT. Recommended protocols of prophylaxis range from mechanical compression devices with early ambulation alone to the addition of chemoprophylaxis.

Thus, it is reasonable for prophylaxis to be guided by deep venous thrombosis, as we did with our patients. Our extended thromboprophylaxis protocol leads to a low incidence of PMVT. At this point, the literature shows insufficient evidence and variability regarding the utility of prophylactic thromboprophylaxis after discharge for the prevention of PMVT, while only three published articles showed the efficacy of extended thromboprophylaxis after LSG.

Despite many advances in bariatric surgery, postoperative venous thromboembolic events remain a challenging issue, with rates ranging up to 2.2%. Chemoprophylaxis is recommended, as these patients are at a higher risk because of the inflammatory and hypercoagulable statuses associated with obesity. 

Nevertheless, a meta-analysis [21] found a similar incidence of PMVT in patients who received prophylactic anticoagulation and those who did not; however, thromboprophylaxis was variable and inconsistent among various studies.

Parikh et al. showed that extended chemoprophylaxis postoperatively could decrease the incidence of PMVT from 0.3% to 0.1%, without a significant increase in bleeding episodes [33]. Between January 2014 and July 2018, there were 13 PMVT patients out of 4228 LSG (0.3%) patients; between August 2018 and March 2019, there was one PMVT patient out of 745 LSG (0.1%) patients. Before July 2018, all patients deemed by the surgeon to be “high-risk” patients postoperatively received extended chemoprophylaxis [5]. After July 2018, all patients who tested positive for thrombophilia postoperatively received extended chemoprophylaxis. A meta-regression analysis [20] showed that patients who received the extended prophylactic anticoagulation protocol had a lower incidence of PMVT than those who received the short-term prophylactic anticoagulation protocol. 

Rottenstereich et al. [20] also compared the incidence after bariatric surgery between patients with and without extended prophylactic anticoagulant. They found no incidence of PMVT (0%) among the 543 patients who were subjected to a protocol of extended prophylactic anticoagulation, and 16 (0.416%) out of the 3843 patients received a protocol of postoperative prophylactic anticoagulation only until the patients’ discharge. 

However, despite the recommendations of the guidelines, the extension of chemoprophylaxis is variable and debatable, as it is often at the surgeon’s discretion, to balance thromboembolic events with the prevention of bleeding [33,34].

Extended chemoprophylaxis is different at different institutions, and it is unclear whether one form of chemoprophylaxis is superior to the others. 

Another question regards whether the chemoprophylactic dose is efficient. Our patient with poor, fulminant evolution was already receiving a maximal dose of LMWH when he developed massive PVT. The main question that should be addressed regards whether the LMWH was effective. Consequently, over the past 5 years, following this dramatic case, we introduced anti-factor Xa (anti-Xa) concentration measurements to monitor the activity of LMWH in all high-risk and difficult cases. 

Our data showed a decreased incidence of PMVT compared with all studies selected and our own data from before we introduced the protocol. Because the literature shows that most PMVT cases appear in the first month, and some of our cases of PMVT had a late onset, we decided at the institutional level in a multidisciplinary team to increase the duration of chemoprophylaxis from 3 weeks to 4 weeks for all patients, irrespective of the risk.

Our new and current protocol for PMVT prevention and therapy is shown in Figure 4.

The limitations of our study are related to the retrospective design and the requirement for further validation in prospective or randomized studies. Nevertheless, we prospectively analyzed data collected in BOLD for a large cohort of LSG patients, who received standardized management and follow up in a high-volume Center of Excellence for Bariatric Surgery. Likewise, our protocol for PMVT prevention and treatment leads to a very low incidence of PMVT and its continuous improvement can still reduce the incidence to zero and improve the outcomes for patients diagnosed with PMVT.

## 5. Conclusions

PMVT is an infrequent but serious complication after laparoscopic sleeve gastrectomy that can result in ischemia with infarction of the bowel spleen or liver.

A history of venous thromboembolic events in obese patients with a hypercoagulable state, predisposing thrombotic factors such as thrombophilia, and postoperative dehydration are the most important risk factors for the development of PMVT after bariatric surgery.

High clinical suspicion of PMVT is required during the postoperative period after LSG. Early diagnosis and especially anticoagulant treatment can produce favorable outcomes. Following the acute event, long-term anticoagulation and monitoring are necessary to prevent PMVT chronic complications and facilitate recanalization.

The prevention of PMVT by a thromboprophylaxis protocol is of extreme importance. Although strong data on PMVT anticoagulant prophylaxis are still lacking, our protocol, with adjusted doses of LMWH and extended prophylaxis for up to 3 weeks, proved efficient. Despite this, we had cases of thrombosis beyond 3 weeks, and we concluded that it was necessary to extend our protocol to 4 weeks for all patients, irrespective of risk. The anti-Xa factor measurements can improve outcomes by identifying the most efficient therapy or prophylaxis levels.

Additionally, screening with a thrombophilia panel for all bariatric patients may be worthwhile due to the high incidence of thrombophilia in bariatric patients (52%) and the high risk of developing thromboembolic events.

## Figures and Tables

**Figure 1 diagnostics-13-00043-f001:**
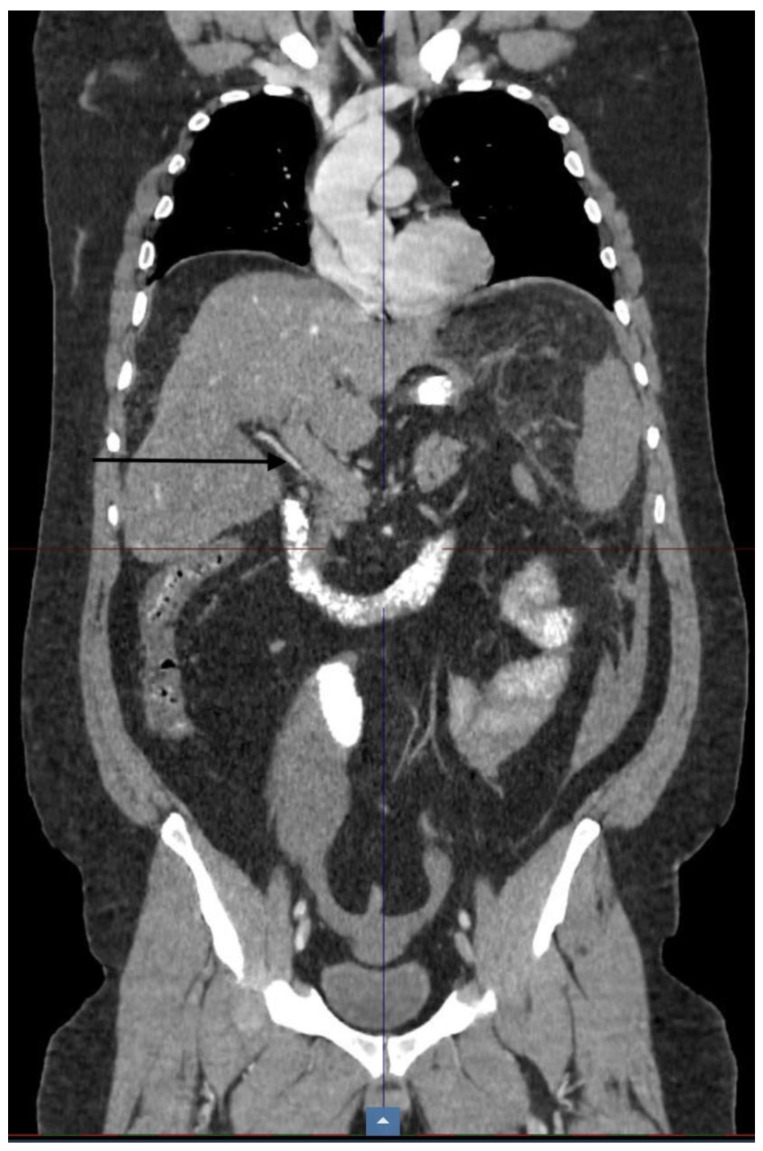
PVT occlusion—CT scan.

**Figure 2 diagnostics-13-00043-f002:**
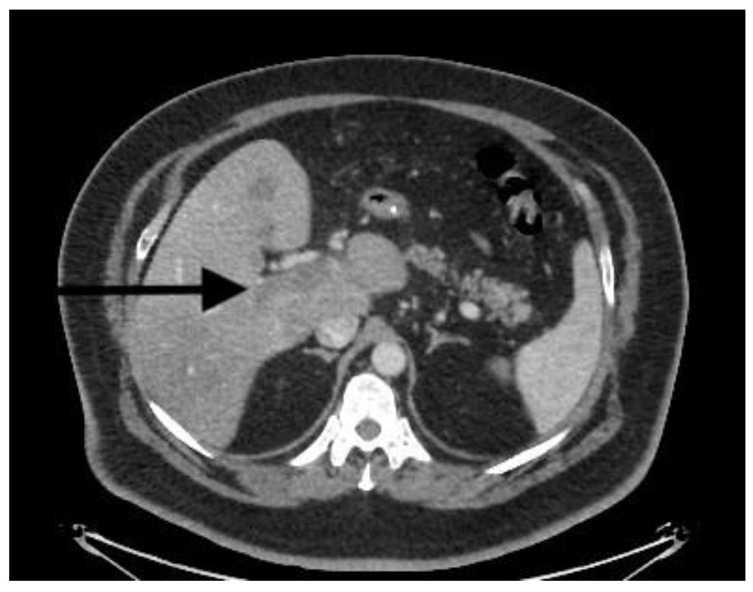
Portal thrombosis CT.

**Figure 3 diagnostics-13-00043-f003:**
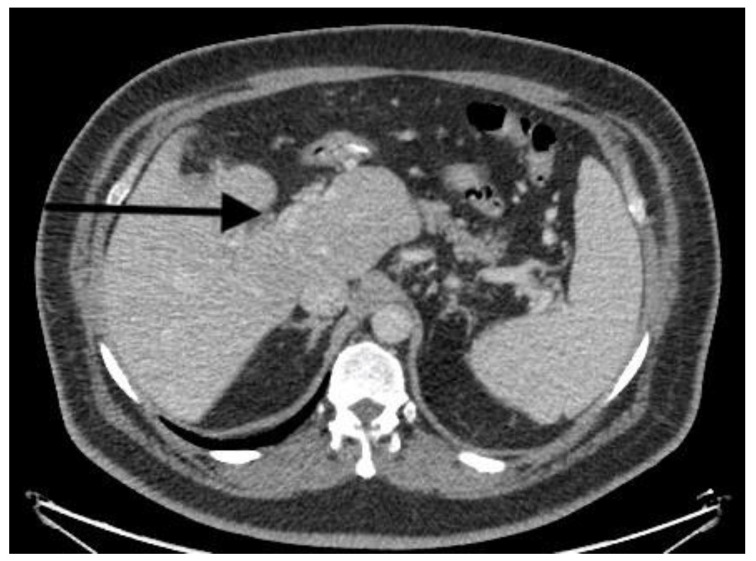
Recanalization of portal vein after 3-month CT.

**Figure 4 diagnostics-13-00043-f004:**
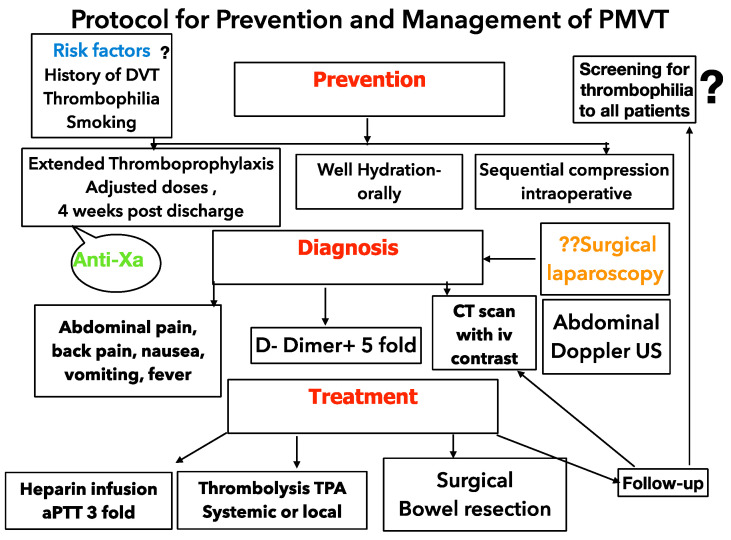
Protocol for prevention and management of PMVT in Ponderas Academic Hospital.

**Table 1 diagnostics-13-00043-t001:** LMWH dose adjustment based on body weight for perioperative DVT/PVT prophylaxis protocol in Ponderas Academic Hospital [14].

Weight	50–100 kg	100–150 kg	Over 150 kg
Dalteparine	5000 UI	5000 UI BID	7500 UI BID
Enoxaparine	40 mg/day	40 mg BID	60 mg BID

**Table 2 diagnostics-13-00043-t002:** The demographics of patients with LSG (2014–2022).

Number of Patients	5154
Age, median (IQR) (years)	40 (31–49)
BMI, median (IQR) (kg/m^2^)	39.4 (34.95–44.55)
Male (n-%)	1565–30.4%
Female (n-%)	3589–69.6%
ASAI (n-%)	413 (8%)
ASA II (n-%)	4000 (77,6%)
ASAIII (n-%)	680(13,1%)
ASAIV (n-%)	61 (1.2%)

BMI—body mass index; ASA—American Society of Anesthesiologists.

**Table 3 diagnostics-13-00043-t003:** Demographics, medical and intraoperative characteristics, diagnosis, treatment, and outcomes of patients with PMVT after LSG in Ponderas Hospital.

Patient	1	2	3	4
Age	29	40	37	55
Sex	M	M	M	M
BMI (kg/m^2^)	50.2	38	44	55,2
Weight (kg)	160	115	127	146
Time of surgery (min)	135	95	60	124
Intra-abdominal pressure (mmHg)	15	15	15	15
History or risk factors of thrombophilia/ DVT	Smoking, hypertension	Double antiplatelet therapy after myocardial infarction and angioplasty	DVT with bilateral PE with anticoagulation–warfarin and antiplatelet therapy	Smoking, hypertension
Days after LSG	12	60	8	34
LMWH prophylaxis for 21 days	Dalteparine 7500 ui sc BID	Dalteparine 5000 ui BID	Enoxaparine 80 mg BID(therapeutic range for 8 days)	Enoxaparine (60 mg BID)
Clinical signs	Abdominal pain, nausea, vomiting	Diffuse abdominal pain (epigastrium) nausea, constipation	Altered general state, severe tachycardia BP100/60, fever, cold sweating, diffuse abdominal pain, tenderness in left and right hypochondrium and lumbar, nausea, vomiting, fulminant evolution to shock, lactate (>15 mmol/L)	Abdominal pain, fever, nausea
Dimers at admission ng/ml	5400	7500	100,000	4400
Doppler ultrasound	Partial PVT	Partial PVT	Complete PVT	Partial PVT
CT scan	Nonocclusive PVT, posterior right branch, and partial superior mesenteric vein	Nonocclusive PVT, right branch, and partial superior mesenteric vein	Extensive occlusive, portal, splenic, and mesenteric thrombosis; ascites evidence of small-bowel hypoperfusion	Partial PVT
Heparin infusion	12 days (aPTT 2,5X)	3 days (aPTT 2,5X)	2 h	3 days, then LMWH for 5 days
Long-term anticoagulation	Apixaban (5 mg BD)	Warfarin (INR)	-	Apixaban (5 mg BD)
Thrombolysis tPA	no	no	no	no
Surgery	no	no	no	no
Hospital stay (days)	13	6	2 h	8
Mortality	no	no	yes	no
Follow-up	Clot regression, no recurrence of PMVT	Clot regression, no recurrence of PMVT	-	Clot regression, no recurrence of PMVT

BMI—body mass index; TPA—tissue plasminogen activator; DVT—deep vein thrombosis.

## Data Availability

Not applicable.

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
