# Peer review of "The Role of Thromboprophylaxis in Patients with Portal Vein Thrombosis: A Life-Threatening Complication after Laparoscopic Sleeve Gastrectomy Following 8 Years of Experience in a Bariatric Center of Excellence"

_diagnostics, 2022, doi:10.3390/diagnostics13010043_

Round 1

Reviewer 1 Report (Previous Reviewer 3)

there's no rebuttal letter

Author Response

I have made again English corrections by MDPI service

All other requirement were met

Thank you very much

Reviewer 2 Report (Previous Reviewer 1)

accept

Author Response

I have made English corrections by MDPI service again

Other requirements have been met

Thank you very much

This manuscript is a resubmission of an earlier submission. The following is a list of the peer review reports and author responses from that submission.

Round 1

Reviewer 1 Report

1- define aim.

2- what is the statisticla method?

(Isik, Arda, et al. "Factors Associated with the Occurrence and Healing of Umbilical Pilonidal Sinus: A Rare Clinical Entity." Advances in Skin & Wound Care 35.8 (2022): 1-4.) and (https://doi.org/10.1089/lrb.2020.0093) suggested studies for the references

Author Response

  • Thank you very much for the comments and suggestions, they are very useful. The manuscript was completely reviewed by professional native English speakers

  • Define aim.

Our study aims to shed light on this relatively rare complication by presenting a series of patients who developed PMVT after LSG, as well as an up-to-date analysis of our database in light of the need to change our protocol for thromboprophylaxis in bariatric patients. We proposed answering two questions: should we perform a thrombophilia workup as a standard practice, and should we extend chemoprophylaxis by more than 3 weeks for all bariatric patients? This study also aimed to look into the possible risk factors and provide a narrative literature review regarding PMVT management and prophylaxis

  1. What is the statistical method?

This answer is provided by the Professor of University of Mathematics who helped me in this study

There is no statistical method involved – no comparisons or tests. We just performed a Kolmogorov-Smirnov test for checking the nomality of continuous data, in order to choose the appropriate way of presenting the data (mean±SD or median and IQR). In the article there is only descriptive statistics used.

The text in the manuscript was changed as following

”Statistical analysis was performed with SPSS version 22 (Chicago, IL, USA). Categorical data were reported as frequencies and percentages, and continuous data as average (mean) and standard deviation, after checking for normality (one-sample case Kolmogorov–Smirnov test). Only descriptive statistics were used.”

We have checked the normality of continuous data with an appropriate statistical test (Kolmogorov-Smirnov) – for the patients group – 5154 patients – which is a very robust one. Due to the fact that the null hypothesis was accepted, we presented the continuous data as mean±SD. For a normally distributed variable, the mean and the median value is almost the same and we don’t see why there is more aprropriate to present the median and IQR instead. The median-approach is suitable for small sample size or for non-normally distributed data

Thank you very much for the support and consideration

Reviewer 2 Report

I read with interest authors' work titled "The Role of Thromboprophylaxis in patients with Portal Vein Thrombosis: a Rare but Life-threatening Complication after Laparoscopic Sleeve Gastrectomy—Case Series and Literature Narrative Review". There are major revisions necessary before publication. However, the authors had made a hard work to summarize some of the immense literature on "sleeve gastrectomy". Therefore, I would like to suggest to focus the review on "LSG comlications". This is obviously the issue of the author. Furthermore, important articles on this topic are lacking. You could consider this paper  doi.org/10.1155/2020/8250904  and discuss it in conclusione section. A language editing service should be ordered for review (native English speak). The written English has to be carefully revised by a native English speaker. 

Author Response

1.Thank you very much for the comments and suggestions, they are very useful. The manuscript was completely reviewed by  professional native English speakers through MDPI services.

2.All answers related to statistics were given  by Professor Raluca Purtan from University of Mathematics  Bucharest who helped me in this study

 3.I reviewed also the conclusions in the light of this rare and potentially fatal complication of PMVT after LSG .Due to the scarce information and thromboprophylaxis  method not standardized and very different  found in literature associated with  lethal potential of this complication,  I would  keep the information obtained from the review , even if this does not cover all studies related to LSG complications, at least I used the most recent meta-analysis and systematic reviews to  strengthen the information from our results.. that recommend extended thromboprophylaxis to reduce numbers of PMVT .. I did not find the doi suggested.. Maybe you give me the full title of the article..

Thank you very much for the support and consideration

Reviewer 3 Report

From a clinical epidemiology and biostats point of view, this manuscript does presents several concerns:

- title, more than case series, it's an observational retrospective trial; hence, I do strongly suggest not to mix it with a narrative review section (they are 2 quite different stories...)

- so, either you report a trial or summarize a narrative review!

- table I, main pts characteristics is totally lacking; who are these guys? Their number is unknown too! Your tables lacks detail

- administered drugs doses and details are totally lacking

- continuous covariates must be described as median(IQR) and not mean/sd

Author Response

1.Thank you very much for the comments and suggestions, they are very useful. The manuscript was completely reviewed by  professional native English speakers through MDPI services.

2.All answers related to statistics were given  by Professor Raluca Purtan from University of Mathematics  Bucharest who helped me in this study

There is no statistical method involved – no comparisons or tests. We just performed a Kolmogorov-Smirnov test for checking the nomality of continuous data, in order to choose the appropriate way of presenting the data (mean±SD or median and IQR). In the article there is only descriptive statistics used.

The text in the manuscript was changed as following

”Statistical analysis was performed with SPSS version 22 (Chicago, IL, USA). Categorical data were reported as frequencies and percentages, and continuous data as average (mean) and standard deviation, after checking for normality (one-sample case Kolmogorov–Smirnov test). Only descriptive statistics were used.”

  1. From a clinical epidemiology and biostats point of view, this manuscript does presents several concerns:

- title, more than case series, it's an observational retrospective trial; hence, I do strongly suggest not to mix it with a narrative review section (they are 2 quite different stories...)

- so, either you report a trial or summarize a narrative review!

Answer: This is true. I have added  ”observational retrospective study” in the method. When I refer to case series I refer to the 4 cases of PMVT found in our group. Also due to the scarce information and thromboprophylaxis  method not standardized and very different  found in literature associated with  lethal potential of this complication,  I would like to keep the information obtained from the review to  strengthen the information and our results.. There are only two more important studies that recommend extended thromboprophylaxis to reduce numbers of PMVT and support our conclusions ( even that those  studies are not standardized  related to the method and duration of thromboprophylaxis) I really hope you understand my point of view related to this pathology because I think clinical aspects are more important than method of design.

  1. table I, main pts characteristics is totally lacking; who are these guys? Their number is unknown too! Your tables lacks detail

Answer: In Table 1 is resumed to our thromboprophylaxis protocol , based on body weight and adapted after recommendation of the UK Hemostasis, Anticoagulation and Thrombosis (HAT) Committee, published on April 2010. It is not related to a specific group of patients under study – is clearly stated in the paragraph preceding the table that the protocol was “applied to all bariatric patients”, starting with 2011.

  1. administered drugs doses and details are totally lacking- I completed doses of prophylaxis therapy in the table 3.
  2. continuous covariates must be described as median(IQR) and not mean/sd

Answer: We have checked the normality of continuous data with an appropriate statistical test (Kolmogorov-Smirnov) – for the patients group – 5154 patients – which is a very robust one. Due to the fact that the null hypothesis was accepted, we presented the continuous data as mean±SD. For a normally distributed variable, the mean and the median value is almost the same and we don’t see why there is more aprropriate to present the median and IQR instead. The median-approach is suitable for small sample size or for non-normally distributed data.

Thank you very much for the support and consideration

Round 2

Reviewer 2 Report

Endoscopic Double-Pigtail Catheter (EDPC) Internal Drainage as

First-Line Treatment of Gastric Leak: A Case Series during

Laparoscopic Sleeve Gastrectomy Learning Curve for

Morbid Obesity. 

https://doi.org/10.1155/2020/8250904

Reviewer 3 Report

The Authors decided not to implement any suggestion, even the absurde mix trail+review, which does counfound the reader. A very low level manuscript, which does remain totally unchanged!!!!!